# Preliminary Evaluation of Bioactive Collagen–Polyphenol Surface Nanolayers on Titanium Implants: An X-ray Photoelectron Spectroscopy and Bone Implant Study

**DOI:** 10.3390/jfb15070170

**Published:** 2024-06-21

**Authors:** Marco Morra, Giorgio Iviglia, Clara Cassinelli, Maria Sartori, Luca Cavazza, Lucia Martini, Milena Fini, Gianluca Giavaresi

**Affiliations:** 1Nobil Bio Ricerche srl, V. Valcastellana 26, 14037 Portacomaro, Italy; giviglia@nobilbio.it (G.I.); ccassinelli@nobilbio.it (C.C.); 2Scienze e Tecnologie Chirurgiche, IRCCS Istituto Ortopedico Rizzoli, Via di Barbiano, 1/10, 40136 Bologna, Italy; maria.sartori@ior.it (M.S.); luca.cavazza@ior.it (L.C.); lucia.martini@ior.it (L.M.); gianluca.giavaresi@ior.it (G.G.); 3Direzione Scientifica, IRCCS Istituto Ortopedico Rizzoli, Via di Barbiano, 1/10, 40136 Bologna, Italy; milena.fini@ior.it

**Keywords:** surface nanolayer, collagen, polyphenols, dental implant, X-ray photoelectron spectroscopy, osseointegration, bone regeneration

## Abstract

To endow an implant surface with enhanced properties to ensure an appropriate seal with the host tissue for inflammation/infection resistance, next-generation bone implant collagen–polyphenol nanolayers were built on conventional titanium surfaces through a multilayer approach. X-ray Photoelectron Spectroscopy (XPS) analysis was performed to investigate the chemical arrangement of molecules within the surface layer and to provide an estimate of their thickness. A short-term (2 and 4 weeks) in vivo test of bone implants in a healthy rabbit model was performed to check possible side effects of the soft surface layer on early phases of osteointegration, leading to secondary stability. Results show the building up of the different nanolayers on top of titanium, resulting in a final composite collagen–polyphenol surface and a layer thickness of about 10 nm. In vivo tests performed on machined and state-of-the-art microrough titanium implants do not show significant differences between coated and uncoated samples, as the surface microroughness remains the main driver of bone-to-implant contact. These results confirm that the surface nanolayer does not interfere with the onset and progression of implant osteointegration and prompt the green light for specific investigations of the potential merits of this bioactive coating as an enhancer of the device/tissue seal.

## 1. Introduction

Remaining challenges for the successful use of implant materials for both the orthopedic and dentistry fields can be tracked back to what could be termed an “original sin” of Branemark’s pioneering studies: his landmark investigation defined principles and protocols for long-term clinical success of osseointegrated titanium screws [1,2,3,4]. In the past, the focus was primarily on titanium/bone interaction, perfecting implant design and surface topography to the present level of efficiency and sophistication, limiting the investigations on bacterial interactions from the main picture. Nevertheless, bacteria claim an increasing role by knocking at the submerged world door as dramatically shown in Figure 1. This is a scanning electron microscope image of the boundary between the cover screw and the implant neck, as it appears after a few weeks service. In short, it is a snapshot of the boundary between the submerged world, buried in bone, and the external, unprotected environment. The impressive bacterial colonization of the emerging portion of the device, even if not in contact with bone, does have a significant impact on osteointegration and the long-term success of the implanted device. These occurrences are bridging the dental field with the emerging realm of osseointegrated percutaneous implants, serving as an alternative method for connecting artificial limbs in amputee patients [5]. Indeed, referring to Figure 1, direct, destructive bacterial colonization of the device in contact with bone in the submerged world of the implant is normally prevented by the soft tissue seal, which should provide a longstanding barrier by allowing the direct attachment of soft tissue to the protruding implant surfaces. However, excluding microbial interactions from the variables affecting the implanted device's lifetime is just an illusion.

Bacteria colonization and biofilm formation on medical devices, unfortunately, represent one of the most feared and challenging complications in orthopedic applications, with an increasing occurrence. In percutaneous implants, among which dental implants are some of the most well-known, the continuous assault of immune response self-defense mechanisms against oral bacteria besieging the submerged implant site results in tissue loss and a continuous downward shift of the boundary. In the majority of cases, marginal bone loss proceeds so slowly that no major functional problem arises within the patient's lifetime. Instead, in genetically predisposed individuals, or following oral microflora dysbiosis, or for a number of not completely understood reasons, the rate of loss of supporting tissue can be so high it may lead to functional failure of the implant well within its time of service [6,7,8,9,10,11]. This is known as peri-implantitis, the major remaining problem of dental implantology. In other types of percutaneous implants, such as bone-anchored hearing devices or direct skeletal attachment systems for prostheses in amputee patients, ensuring secure and stable epithelial cell adhesion to the implant surface is crucial for preventing the onset of both superficial and deep infections. In these specific applications, failure to achieve proper skin sealing may result in the ‘marsupialization phenomenon’ or ‘permigration’, potentially forming pockets where waste material can accumulate, providing an ideal environment for bacterial growth and proliferation [12].

For these reasons, ensuring a stable and proper seal between the implant surface and host tissue to prevent peri-implantitis or pocket formation is the main goal of next-generation implants and a hot topic in today’s research on percutaneous implantology devices [12,13].

Among strategies to prevent the onset of inflammation and infection, an interesting avenue of research involves upgrading the surface chemistry of titanium [14,15,16]. The underlying rationale is to impart to the Ti device, through biochemical signaling from surface-linked molecules, better healing properties, leading to an enhanced response to bacterial challenges and ensuing the side-effects of self-defense-related inflammatory mechanisms [17].

We are focusing on the surface treatment of titanium implant systems and on the development of novel technologies to control tissue response at the titanium–host tissue interface. In particular, we are pursuing the exploitation of anti-inflammatory and antibacterial properties of a class of molecules from the plant kingdom, i.e., polyphenols [18,19]. Phenolic phytocompounds are well-known antioxidants, possess antimicrobial properties [20,21,22,23,24,25], and can be linked to material surfaces through a number of different techniques [26]. The interest in biomedical applications of polyphenols is continuously rising, both through the surface linking of purified molecular species, so-called monoligand systems, and the use of multi-molecular phenolic mixtures obtained by infusion or extractions from tea, grape seeds and skins, or other vegetal sources, known as multiligand systems [27].

The present work is part of a project aiming to upgrade the surface chemistry of titanium implants by building up a surface layer made of anti-inflammatory, antimicrobial, multiligand polyphenols from pomace extracts. In particular, we have recently shown, in a submitted article under revision, that a multiligand-polyphenol-rich extract from winery by-products shows the inhibition of growth of G+ *Streptococcus mutans* and G- *Porphyromonas gingivalis*, two bacteria strains of relevance to biofilm formation especially in the oral cavity [28]. The same extract modulates inflammatory response in in vitro tests involving murine macrophages. While these properties are consistently demonstrated in the case of the relevant molecules in solution, it is not known yet whether and how much linking them to a substrate surface will keep relevant activities. Our present research efforts aim to investigate surface-engineering processes that could impart to the Ti device surface the same properties as those shown by the multiligand polyphenols extracted in the solution.

Based on previous studies, our preferred strategy is to build up a specific interfacial environment through a multilayer approach [19]. For this reason, besides polyphenols, we use collagen both as a surface primer to enhance the yield of polyphenols coupling to the Ti surface and because of its own pro-healing properties both on soft and hard tissue [29,30]. Coupling the antibacterial and anti-inflammatory properties of polyphenols to enhanced tissue interaction provided by collagen is the ultimate aim of the present research project on the quest for inflammation/infection-preventing implants.

Our approach involves the application of three subsequent layers, namely collagen/polyphenols/collagen. This allows for maximizing the surface density of relevant molecules without building a too-thick surface layer that could be mechanically weak in adhesion and cohesion.

In this respect, to build up a sound basis for dedicated tests, we try to answer two fundamental questions of great relevance for practical clinical application, namely:-The first point involves the actual structure of the surface layer. The driving force for building multilayers involving polyphenols and collagen is not just that of charge interaction, as in most conventional layer-by-layer (LBL) systems [31]. It is well known that polyphenols act as chemical cross-linkers of collagen, as exploited in leather tanning for centuries [32]. Thus, we should not expect to build a simple layered structure on the surface; rather, the different layers will interplay according to relevant chemistry. The first aim of this work is to try to understand the arrangement of collagen and polyphenols on top of titanium through surface analysis.-Then, the surface layer is intended to enhance the device response at the “battlefront” shown in Figure 1. Thus, from a practical point of view, in a production setting, it is much more convenient to apply it to the whole implant surface, confined to a reaction in the liquid environment to a specific area rather cumbersome and expensive. It is then mandatory to evaluate the effects of the surface layer on conventional osteointegration to ascertain that it is not delayed or impaired by the chemistry of the layered structure.

This work tries to respond to the two previous questions by following, through X-ray Photoelectron Spectroscopy (XPS), the building up on the titanium surface of a layer obtained through the combination of polyphenols-rich extract and collagen. Then, we present the results of in vivo bone implant studies at two time points in a rabbit model. The latter involves titanium fixtures featuring both the main topographies presently adopted for dental implants, which are machined and used in tissue-level implant necks, and microrough, obtained through acid etching and used in bone-level implant necks.

## 2. Materials and Methods

### 2.1. Materials

All chemicals were analytical reagent grade. Acetone, acetic acid, Folin–Ciocalteu reagent, 2,2-Diphenyl-1-picrylhydrazyl, sodium carbonate, and gallic acid were purchased from Sigma-Aldrich (St. Louis, MI, USA). Collagen type-1 from a bovine source was purchased from Symatese (Chaponost, France). Dulbecco’s phosphate-saline buffer (DPBS) was purchased from Gibco Invitrogen (Cergy-Pontoise, France). Water was ultrapure and obtained through Milli-Q Advantage A10 (Merck, KGaA, Darmstadt, Germany. Red grape pomace was purchased from a local wine producer (Croatina grape from ALEMAT, Penango, AT, Italy).

### 2.2. Preparation of the Polyphenols Rich Pomace Extracts (PRPE)

The pomace was stored as received from the winery at −20 °C under vacuum. PRPE was obtained by drying the pomace at 37 °C ± 5 °C, grounding it in a bladed mill (GM 200, Retsch), and washing it with acidified water. The grape pomace was extracted using an automatic extractor (Micro C TIMATIC, Spello (Pg), Italy). Three hundred grams of winery by-products were soaked in 2000 mL of 50:50 acetone:water (*v*/*v*). The extraction cycle is fully automatic and alternates a dynamic phase with a static phase, which, thanks to the programmable recirculation, ensures a continuous flow of solvent to the interior of the plant matrix.

Aqueous extract was obtained to eliminate the acetone under reduced pressure in a rotavapor. The concentrated extract was separated by centrifugation (7000 rpm, 5 min), and the supernatant fraction was used to treat the Ti disk.

Aqueous extract characterization is reported as Appendix A.

### 2.3. Preparation of Ti Samples for XPS Analysis and In Vivo Study

Ti disks for XPS analysis: Grade 4 Ti disks 8 mm in diameter and 1 mm in thickness were used for the XPS studies. Ti disk samples used for XPS analysis were left-machined to not add further variables to the angle-dependent XPS analysis.

The Ti disks were subjected to solvent-cleaning techniques and a final step of air-plasma glow discharge cleaning (Ti samples).

The preparation of samples involves the adsorption of fibrillar collagen to the Ti surface by soaking the machined Ti disk for 2 h in a collagen solution 0.3% *v/v* diluted until 0.15% *v/v* with DPBS at 37 °C. Collagen-adsorbed Ti disks are coded as Tic. After soaking in collagen/DPBS solution, the Ti disks were rinsed with ultrapure water three times. The second step of functionalization consisted of soaking the Tic disk in the aqueous extract for 2 h at room temperature (Ticp) and then rinsing three times in ultrapure water. In the third step of functionalization, Ticp disks were soaked overnight at 37 °C in a 0.3% collagen solution diluted to reach a concentration of 0.15% *v/v* using DPBS, again to obtain collagen fibrillation on the surface (Ticpc), sample after soaking process were rinsed three times in ultrapure water and left to dry under a fume hood at room temperature.

Ti screws for in vivo studies: Titanium Grade 4 screws, 5 mm length and 2 mm diameters, were used for bone implant studies. One set of Ti samples was subjected to conventional Double Acid Etching (DAE) treatment to provide the typical microroughness adopted in state-of-the-art dental implants (Appendix A). Another set was left-machined. Both sets of Ti samples were subjected to solvent cleaning techniques and a final step of air-plasma glow discharge cleaning.

Four different surface conditions have been used in animal implants:-Ti-machined (Ti mach), which represents the control surface of the in vivo study;-Ti-etched (Ti rough);-Ti-machined functionalized with all three steps of functionalization (Ticpc mach);-Ti-etched functionalized with all three steps of functionalization (Ticpc rough).

Fifteen samples for each condition have been produced. Ti mach, Ti rough, Ti cpc mach and Ticpc rough samples were packed using medical paper in an ISO7 clean room and sterilized by Gamma irradiation (25 kGy).

### 2.4. X-ray Photoelectron Analysis of Treated Disks

X-ray photoelectron spectroscopy (XPS) analysis was performed using a Perkin Elmer PHI 5600 ESCA system (PerkinElmer Inc., Waltham, MA, USA). The instrument has a monochromatized Al anode operating at 10 kV and 200 W. The base pressure was maintained at 10^−8^ Pa. The diameter of the analyzed spot is approximately 300 μm; the analyzed depth is about 8 nm, keeping the angle between the electron analyzer and the sample surface at 45°.

To gather information on the vertical distribution of elements within the surface layer, Angle-Resolved XPS (ARXPS) was performed. Varying the emission angle at which the electrons are collected enables electron detection from different depths [33]. ARXPS provides information about the thickness and composition of ultra-thin films. In ARXPS experiments, we used sample tilting angles of 15, 45 and 75 degrees, providing approximated sampling depth of 4, 8 and 10 nm.

Analysis was performed by acquiring wide-range survey spectra (0–1000 eV binding energy) and detailed high-resolution peaks of relevant elements. The elements were quantified using the software and sensitivity factors supplied by the manufacturer. High-resolution C1s peaks were acquired using a pass energy of 11.75 eV and a 0.100 eV/step resolution. C1s peaks were referenced to the internal standard C-C component at 285.00 eV.

### 2.5. In Vivo Study

The in vivo study was performed following Italian law on animal experimentation (Law by Decree 26/2014) after the approval of the Italian Ministry of Health (Project authorization n. 81/2022-PR9). The manuscript for this section has been written to adhere to the ARRIVE guidelines, with the ARRIVE Essential Checklist provided as a Appendix A. According to the prior power analysis calculated using the G*Power software v.3, the sample size for the study was *n* = 12 animals, with 6 per experimental time point (2 and 4 weeks). An effect size of f ≥ 0.50 was hypothesized for the quantitative histomorphometric evaluations based on previous results [34,35] in a two-way ANOVA model (device and experimental time), with 1−β = 0.80 and α = 0.05. Twelve male New Zealand rabbits were sourced from an authorized supplier (Charles River, SAS France RMS) weighing 2.63 ± 0.16 kg, which were individually housed in cages with enriched materials and provided with standard long-term maintenance pelleted feed (Mucedola S.R.L., Settimo Milanese—Milano, Italy), along with ad libitum access to water.

After the quarantine period, the animals underwent a surgical procedure for implant placement. Premedication involved intramuscular injection of 44 mg/kg ketamine (Imalgene 1000, Merial Italia S.p.A., Assago-Milan, Italy) and 0.3 mg/kg xylazine (Rompun 25 mL, Bayer S.p.A., Italy). General anesthesia was induced and maintained by administering a gas mixture of O_2_/air (60%/40%) and 2–3% of sevorane. Under sterile surgical conditions, following trichotomy and disinfection of the hind limb surface, bilateral longitudinal cutaneous and subcutaneous incisions were made on the lateral surface of the knee until reaching the muscle layer, exposing the lateral distal femoral condyle surface. After periosteal removal, low-speed perforation of the bone tissue was performed with tips of different diameters, accompanied by continuous irrigation with room temperature saline solution, resulting in a bone loss of 2 mm in diameter and 5 mm in depth. Employing the same surgical approach but with a medial access route, bone defects of the same dimensions were created at the level of the medial surface of the proximal tibia. Implant randomization was conducted for each experimental time point by assigning n = 3 femoral sites and n = 3 tibial sites for each implant surface. This ensured that each animal received all four implant surface conditions., Following material implantation, the surgical wounds were sutured layer by layer.

Postoperatively, animals received analgesic and antibiotic therapy, comprising: (1) intramuscular injection of 1.5 mg ropivacaine hydrochloride (Ropivacaine Kabi 7.5 mg/mL—Fresenius Kabi Italia Srl, Isola della Scala VR) at the end of the surgical procedure; (2) application of a 1/3 fentanyl-based transdermal patch (MATRIFEN 50 µg/hour—Grunenthal Italia Srl, Milan, Italy) and intramuscular injection of 50 mg/kg/day for 3 days of sodium metamizole (FARMOLISINA, Vetem SpA, Porto Empedocle—Grosseto, Italy); and (3) intramuscular injection of 0.5 mg/kg flumequine (FLUMEXIL, Fatro S.p.A. Ozzano Emilia—Bologna, Italy) for 3 days. During the postoperative follow-up, daily examinations were conducted for the initial 5 days post-surgery, followed by weekly visits throughout the study’s duration (2 and 4 weeks). The animal welfare evaluation involved monitoring general and local conditions such as the mobility and functionality of the operated limbs, weight, major organ functions, food, and water consumption. If weight loss, infection, or surgical wound lesions were observed, the pharmacological and/or analgesic protocol was adjusted. Humane endpoints were established in advance: body weight loss greater than 20%, severe limb injuries, fractures, or significant alterations in major organ functions.

During the materials surgical implantation and retrieval phase, as well as histologic and histomorphometric investigations, codes were assigned to the materials to ensure that the researchers involved were blinded to the different surface treatments.

#### 2.5.1. Bone Sample Harvesting and Processing

At the end of each experimental time, under general anesthesia as previously described, the animals were pharmacologically euthanized with intravenous injection of 1 mL Tanax (Hoechst AG, Frankfurt-am-Mein, Germany). Joints were macroscopically evaluated to detect any signs of alterations (hematoma, edema, inflammatory and/or infectious reactions) present during harvesting. Subsequently, femurs and tibias from both joints were harvested and fixed in 4% paraformaldehyde buffer for 48 h. Following the fixation step, samples were washed in distilled water and dehydrated in increasing solution of ethanol concentration up to ethanol 100% to completely remove the aqueous component from the tissues and allow penetration of the embedding medium represented by methylmethacrylate-based solutions (Methacrylate; Merck, Shuchardt, Hohenbrunn, Germany). Upon resin polymerization, samples were cut along a frontal cutting plane using a Leica SP1600 diamond blade microtome (Leica Microsystems Srl, Milan, Italy). The central sections of each implant were thinned and polished using abrasive papers with a Saphir grinding system (Saphir 550, ATM GmbH, Mammelzen, Germany) until obtaining histological sections with a final thickness of approximately 15 ± 10 µm. After the grinding and polishing process, the sections underwent histological staining with Stevenel’s Blue counterstained with Fast Green (Figure 2A). Histological preparations were acquired with an Aperio Scanscope CS System high-resolution digital pathology scanner (Leica Biosystem Imaging, Nussloch, GmbH) or with an optical microscope (Olympus BX51, Olympus Italia, srl., Milano, Italy).

For each histological section, a Region of Interest (ROI) was identified, on which the percentage of contact between the material surface and bone tissue (Bone to Implant Contact—BIC, %) was measured using ImageJ software (National Institutes of Health, Bethesda, MD, USA) (Figure 2B). First, the perimeter in the vertical direction of three consecutive threads was identified on both the cranial and caudal sides, starting from the first complete thread of the implant up to the beginning of the fourth thread. This length was used to determine the value of BIC, as shown in Figure 2C, expressed as the ratio between the material perimeter and the perimeter of bone tissue/material surface within the ROI.

#### 2.5.2. Statistical Analysis

The statistical analysis was performed using R software, version 4.2.2 (R Core Team. 2022. R: A language and environment for statistical computing. R Foundation for Statistical Computing, Vienna, Austria). After confirming the non-normal distribution of the results (Shapiro–Wilk test) and the heterogeneity of variance, the analysis proceeded using the non-parametric Kruskal–Wallis test followed by the Dunn test for multiple comparisons with *p*-values corrected using the Sidak method. Since the Kruskal–Wallis test does not allow for a two-way analysis (material and experimental time), a non-parametric analysis was conducted using the interaction of the two factors, ‘material’ and ‘experimental time’, as a factor (independent variable). The data are reported as the Median and Interquartile Range (IQR) at a significance level of *p* < 0.05.

## 3. Results

### 3.1. XPS Analysis

Table 1 shows the results of the XPS analysis obtained at the conventional take-off angle of 45 degrees. The relevant survey XPS spectra are included in Appendix A. The Ti surface shows values in agreement with those expected from clean Ti surfaces, resulting from the combination of the Ti oxide surface layer and some carbon contamination due to ubiquitous hydrocarbon adsorption. Collagen adsorption in the first step is confirmed by the significant increase of the surface N and C concentration. Some Ti is still detected on the Tic surface—the adsorbed layer is thinner than the XPS sampling depth. Following polyphenol coupling to the collagen-coated Ti surface, as expected, the O concentration increases and the N concentration decreases. Ti is now barely detectable, suggesting that the thickness of the surface layer is close to the XPS sampling depth. In the last step, Ti is no longer detected, and the N signal increases again due to the further contribution of the N-containing functionalities to the peptide bonds of collagen.

Interesting indications are supplied by the angle-resolved analysis shown in Table 2. The Tic data show a gradient in the Ti concentration as the sampling depth increases, confirming that the surface is vertically not homogeneous, with the thickness of the adsorbed collagen layer thinner than the XPS sampling depth. Moving to Ticp, the grazing angle (15 deg) data now show that the first few nanometers of the surface are completely organic in nature, while some Ti signal is still detected, pushing the sampling depth to about 10 nm. The growth of the surface organic layer is confirmed by Ticpc data: Ti is no longer detected, and along the XPS sampling depth, some limited fluctuation of the N/O ratio is observed.

Analysis of the C1s photoelectron peak allows us to obtain information on the chemical environment built on the sample surface, as shown in Figure 3. The C1s peak of Tic shows the hallmarks of proteinaceous materials, with a significant component due to the amide bond O=C-N. The chemical environment of carbon changes completely after polyphenol adsorption: the amide component becomes a minority in terms of the 286.5 eV C-O component. Obviously, there are a significant amount of phenolic and carbon-oxygen single bonds contained in the polyphenols’ molecular structure. Interestingly, after further collagen adsorption, the Ticpc C1s peak does not show a new increase of amide component; instead, a broad, poorly resolved peak occurs. This is an actual feature of C1s photoelectrons and not an artefactual broadening of the peak due, for instance, to charge accumulation caused by the increasingly insulating nature of the surface layer. Less convoluted peaks of other elements, such as O1s and N1s, keep the Full Width at Half-Maximum (FWHM) shown on Tic and Ticp. In short, the topmost collagen layer does not show a distinct, proteinaceous identity, as in the case of the Tic one. Rather, the peak shape suggests a mix of closely spaced carbon functionalities, likely arising from polyphenol–collagen chemical interactions.

### 3.2. In Vivo Studies

The animals exhibited good tolerance to the surgical implantation procedures, and no local or systemic complications were observed throughout the experimental period. Recovery from anesthesia was rapid, and during the postoperative course, no adverse effects or tissue responses related to the implant procedure or materials were observed either locally (such as redness and/or swelling of the skin) or systemically. The macroscopic evaluation of the joints, performed before the bone segment’s explant to identify potential tissue reactions, did not reveal the presence of hematoma, edema/swelling, infection, or any other particular responses related to the experimental materials, neither at 2 nor 4 weeks. To avoid any bias related to treatment, each sample was identified using the same identification code assigned to the animal, which was individually caged, specifying the anatomical site of origin (right or left femur, right or left tibia).

Histological slides were prepared and analyzed for six surface-type materials at each experimental time point, with some deviations: at 2 weeks, two cases from the Ticpc match group and one case from the Ti match group were excluded due to implant placement at the diaphysis level, and at 4 weeks, another case from the same Ti match group was excluded for the same reason.

The histomorphometric results of the BIC measurement are shown in Figure 4. At the early phase of 2 weeks, no significant differences were detected among the four different tested surfaces. Qualitatively, rough surfaces showed higher, even if not statistically different, figures of BIC in each set of samples. Also, the Ti set seems slightly higher than the Ticpc set. At 4 weeks, there is a trend towards the superiority of rough surfaces over machined surfaces, which was statistically significant in the case of Ti. The two rough surfaces show the same mean BIC value. Histologically, two weeks after the implant surgery procedures, the presence of newly formed tissue was observed, not only of a connective nature, rich in vascular units and fibroblasts, but also characterized by newly deposited bone tissue (woven bone). Notably, the osteoid tissue is detected, particularly near the implant surfaces, and is associated with intense osteoblastic activity and numerous vascular units. Noteworthy is the absence of fibrous tissue formation or significant accumulation of inflammatory cells around any type of implant (Figure 5).

At four weeks, the woven bone has undergone partial replacement with more organized trabecular bone, incorporating the presence of mature lamellar bone. Indeed, by four weeks, the maturation process of the bone tissue becomes more pronounced: the mineralized bone matrix directly interfaces with the implant surfaces, and the presence of osteoid tissue, along with active osteoblasts, still indicates ongoing regenerative processes. Even at four weeks, there are no noteworthy accumulations of inflammatory cellular elements (Figure 6).

## 4. Discussion

Data arising from bone implant studies, especially those related to the BIC parameter, confirm how an appropriate roughness can positively influence the initial phases of the bone tissue/material surface interaction and the stability of the implant itself. These data reflect the basis of present-day implantology: a physical attribute, which is surface nano- and microroughness, drives bone response to implants and provides enhanced properties, in particular faster secondary stability, concerning machined surfaces. As depicted in classic papers by Davies et al. [36], this occurs because of the enhancement of contact osteogenesis on microrough surfaces. This attribute is largely independent of titanium chemistry; hiding the surface titanium oxide layer, as in the present case, does not prevent roughness to rule bone response.

Besides existing knowledge on stimulation of osteointegration by surface roughness, it is understood that the main weak link for the long-term success of implants is not the stability of the bone–implant interface but bacteria-promoted infection/inflammation. The latter results in peri-implant tissue destruction due to “friendly fire” from the host defense system response to bacteria [6,7,8,9,10,11]. It follows that counteracting bacterial adhesion and promoting healthy tissue healing and sealing are important clinical requests of implant materials.

In the present work, we attempted to build a surface layer based on polyphenols, an exceedingly interesting class of phytocompounds showing antioxidant and antimicrobial properties [18,19,20,21,22,23,24,25,37,38]. Irreversible, adsorbed preteinaceous collagen at the liquid–solid interface [39,40,41] was used as a primer to enhance the adhesion of phenolics to titanium and because of its known effects on tissue healing. Our first aim was to gather information on the chemical nature and thickness of the surface layer obtained by the sequential adsorption of collagen and polyphenol-rich extract.

The obtained data provide significant information. In particular, the thickness of the surface layer increases by a few nanometers at each step. It is close to the XPS sampling depth using a 15 deg grazing angle as the first layer is adsorbed; it is close to the XPS sampling depth using the take-off angle of 45 degrees, roughly 8 nm when the second layer is applied. It fills all the thickness of the 75-degree sampling depth by applying the third layer.

The chemical nature of the surface layer is well reflected in C1s peaks obtained by XPS (Figure 2). In particular, the first and second layers show the expected features of amide-functionality-rich proteinaceous collagen and carbon-single-bond-oxygen-rich polyphenols. The C1s peak after the last step, instead, is less defined, and it results from the convolution of many closely spaced carbon functionalities based on single and multiple carbon–oxygen and carbon–nitrogen bonds. This evidence confirms that the driving force for surface assembly is not just charge interaction, but chemical interactions occur. Indeed, the cross-linking of collagen by polyphenols has been widely investigated and applied in many technological domains [32,38]. To stay in medical applications, polyphenolic compounds and mixtures are used to cross-link the collagen of dentine to increase the mechanical strength of the adhesion layer between dentin and composites [42,43,44,45,46,47]. Proanthocyanidins (PA), which make up a significant fraction of the extract we used, have been shown to induce effects both to the secondary and tertiary structure of collagen, and a significant role for hydrogen bonding in the stabilization of collagen by PA has been suggested [48,49,50]. Besides hydrogen bonding, true covalent interactions have been detected by infrared spectroscopy between PA and collagen [51,52]. In particular, they occur through auto-oxidation of catechol moieties to ortho-quinone groups, available for Schiff base formation with free amine groups on proteins. The broadening of the C1s peak is consistent with introducing a further component due to the imine (C=N) bond of the Schiff base, located between the carbon single-bond oxygen and carbon double-bond oxygen [53] and partly overlapping with them.

As to in vivo bone implant studies, two different experimental timepoints were investigated, 2 and 4 weeks. The comparatively short 2-week time point was chosen to detect the potential slowing-down of osteointegration in the early phases, which is likely more sensitive to differences in surface chemistry and of great clinical relevance. The 4-week time point represents a more mature stage of osteointegration, and it should reflect the complete outcome of the device/bone interaction. However, our observations and measurements focused solely on the response of the trabecular bone tissue. Further studies are needed to understand the response at the cortical bone level, as different features characterize the trabecular and cortical bone structure.

Regarding the polyphenolic component, the in vivo evidence about implant functionalization with polyphenols extracted from grace pomace seems to confirm the osteogenic and anti-inflammatory properties associated with these molecules. No significant inflammatory reaction was observed at 2 or 4 weeks after implant related to the functionalized surfaces, confirming their safety profile.

Evidence exists that collagen can enhance the BIC of titanium microrough surfaces in the same animal model used in the present investigation at the experimental times. In contrast, in the present case, no effect was detected [54,55]. This could suggest that the molecular signaling involving interfacial new bone formation could have been muted by conformational and chemical changes occurring in the collagen molecules on interactions with polyphenols. The dedicated tests built on present findings will try to understand the structure–function effects on activities specifically aimed at counteracting bacterial inflammation/infection.

## 5. Conclusions

In conclusion, the experimental activity performed shows that:

Using a multilayer approach, a mixed, multiligand polyphenols-collagen layer is built on top of titanium surfaces. The thickness of the layer is about 10 nm. The chemical nature of the surface layer is not just dictated by simple physical, charge-driven adsorption of the subsequent layers. Rather, in agreement with literature reports, evidence of chemical interactions between polyphenols and collagen, leading to covalent bond formation, is suggested by the study of the carbon chemical environment, as captured by XPS analysis. Short-term bone implant studies in rabbits indicate that the surface organic layer does not affect the path and the amount of peri-implant bone formation, as evaluated by BIC. Contact osteogenesis is enhanced by the physical attribute of roughness, leading to enhanced BIC of microrough surfaces vs. the machined surfaces, independently from surface chemistry.

The present results open the way for specific studies on the effect of the polyphenol–collagen coating on the mechanisms of relevance to enhance the antibacterial properties of titanium bone implants.

## Figures and Tables

**Figure 1 jfb-15-00170-f001:**
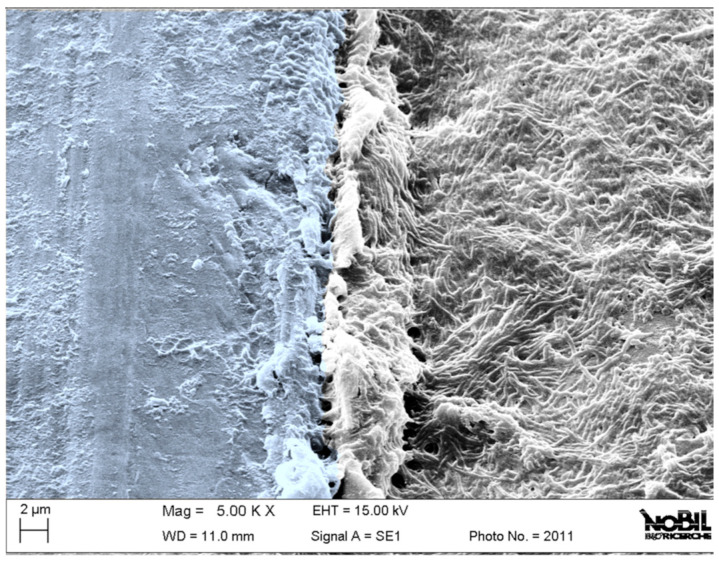
Scanning electron microscope image of a functionally working retrieved dental implant after a few weeks from implantation. The image shows the boundary between the cover screw (right), fully covered by bacteria thriving in the oral cavity, and the submerged portion of the implant (left), protected from bacterial colonization by the soft tissue seal.

**Figure 2 jfb-15-00170-f002:**
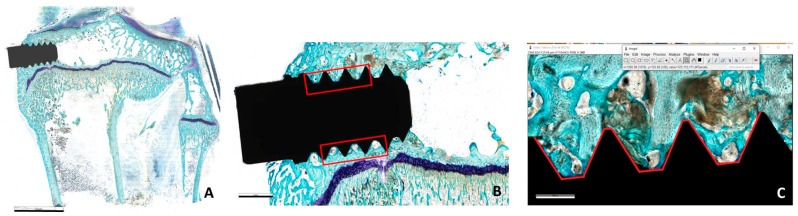
(**A**) A representative image of the implant site in the trabecular bone tissue 4 weeks after surgery (magnification 0.4×, scale bar: 5 mm); (**B**) the selected ROI where histomorphometric parameters were measured (magnification 2.1×, scale bar: 1 mm); (**C**) the profile of the screw threads were BIC parameter is shown in red, and was measured surrounded by trabecular bone tissue (magnification 8.4×, scale bar: 300 µm). All images were acquired with an Aperio Scanscope CS System digital scanner after staining them with Stevenel’s Blue/Fast Green.

**Figure 3 jfb-15-00170-f003:**
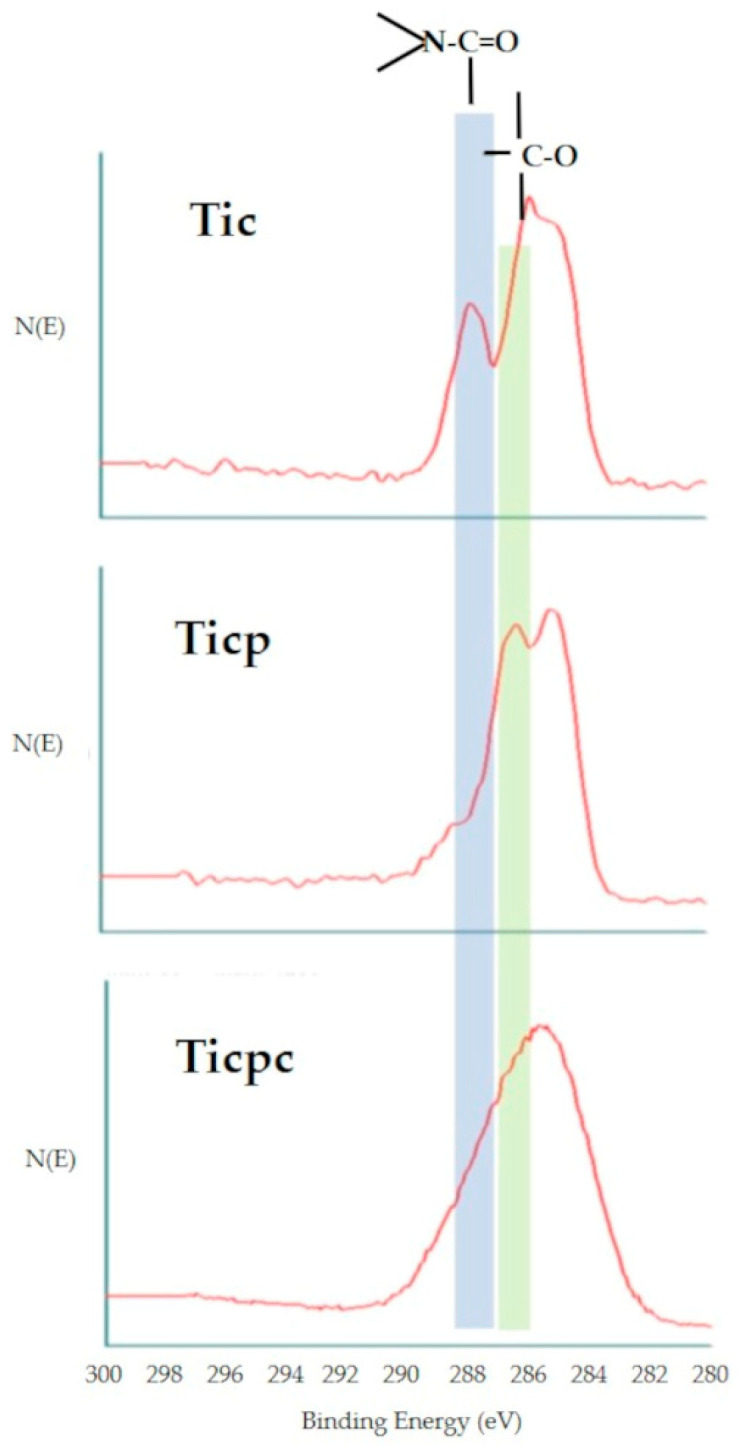
C1s core peaks obtained by XPS analysis of TiC, Ticp, and Ticpc. All peaks reference the 285 eV component of the C-C and C-H bonds.

**Figure 4 jfb-15-00170-f004:**
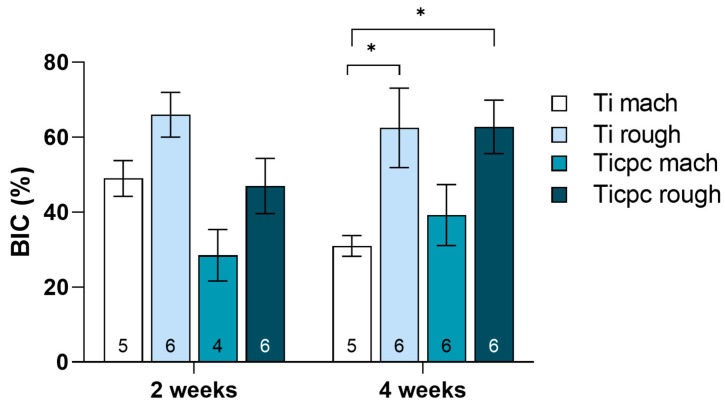
Histogram representing the values obtained for BIC histomorphometric parameters (*n* for each experimental group was reported at the bottom of the histograms). The statistical analysis for the BIC parameter (Kruskal–Wallis χ^2^ = 19.83, *p* = 0.006) did not show statistically significant differences at 2 weeks among the different surfaces. At 4 weeks, a significant difference was observed between Ti mach surfaces versus Ti rough and Ticpc rough B (*, *p* = 0.016).

**Figure 5 jfb-15-00170-f005:**
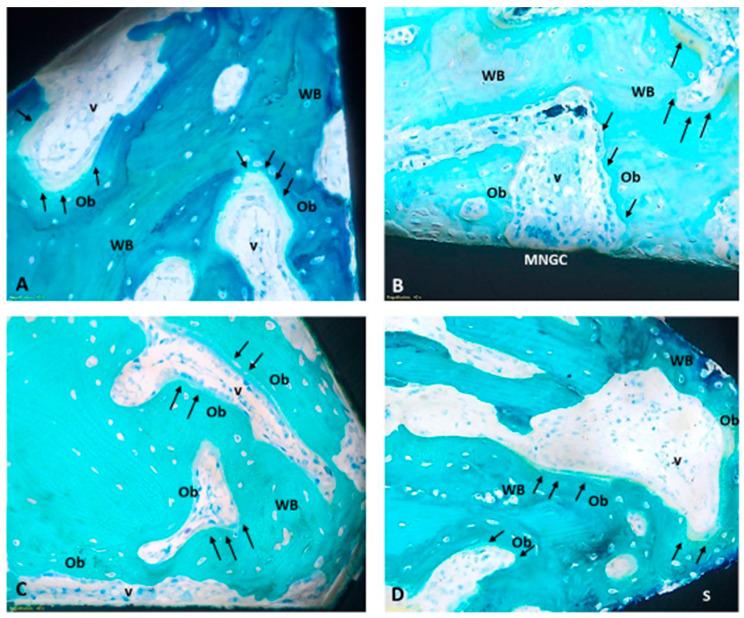
Representative panel of the histological findings observed for the different surface materials (**S**) at two weeks. (**A**) Ti mach; (**B**) Ti rough; (**C**) Ticpc mach; (**D**) Ticpc rough). Within the screw coils, the presence of newly formed vessels (**v**) and newly deposited tissue (woven bone, **WB**) covered by rows of actively depositing osteoblasts (**Ob**) has been identified, as indicated by the presence of osteoid matrix (**black arrows**). No significant accumulations of inflammatory cells around the materials were noted; some multinucleated giant cells (**MNGC**) were sporadically observed, and there was no fibrous tissue interposed between the implant surface and the bone tissue. (Stevenel’s Blue/Fast Green staining; images acquired Optical Microscope Olympus BX51 at 40× magnification).

**Figure 6 jfb-15-00170-f006:**
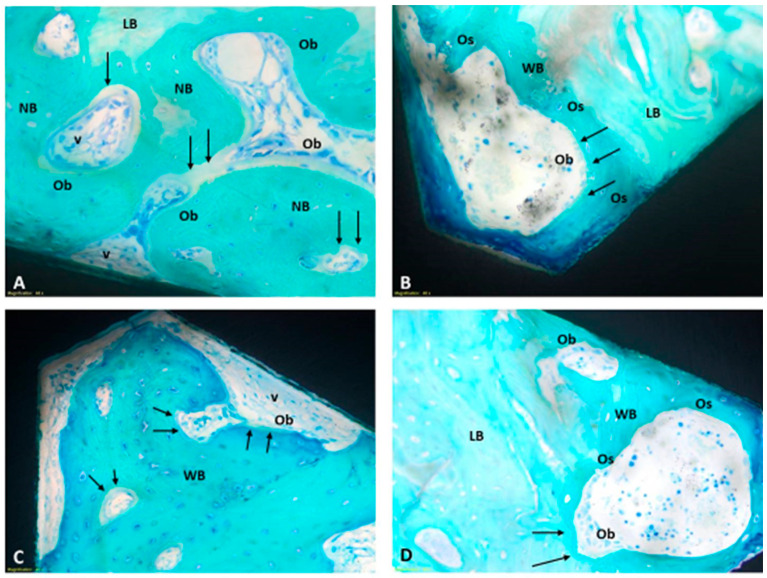
Representative panel of the histological findings observed for the different surface materials at four weeks. (**A**) Ti mach; (**B**) Ti rough; (**C**) Ticpc mach; (**D**) Ticpc rough). Some of the presented images exhibit a distinct chromatic intensity in the bone tissue due to the Fast Green staining: a pale stain for mature, lamellar bone tissue (**LB**) and a vibrant stain for newly deposited tissue (**NB**). All images provide details.

**Table 1 jfb-15-00170-t001:** Surface composition (% at.) of Ti disks as detected by XPS after the different surface modification steps.

Sample	O	Ti	N	C
**Ti**	48.4	19.5	0.4	31.7
**Tic**	22.5	1.7	13.9	61.9
**Ticp**	26.4	0.4	5.2	68.0
**Ticpc**	21.6	-	11.8	66.5

“-”: under the detection limit - no element detected.

**Table 2 jfb-15-00170-t002:** Results of angle-resolved XPS measurement of Ti disks' surface composition (% at.) after the different surface modification steps. The sampling depth increases from 15 to 75 degrees, as explained in the text.

Take Off Angle	Sample	O	Ti	N	C
15	Tic	18.2	0.8	12.8	68.2
45	Tic	22.5	1.7	13.9	61.9
75	Tic	26.0	3.2	14.0	56.8
15	Ticp	24.9	-	4.9	70.2
45	Ticp	26.4	0.4	5.2	68.0
75	Ticp	28.4	0.8	5.5	65.3
15	Ticpc	20.6	-	13.1	66.2
45	Ticpc	21.6	-	11.8	66.5
75	Ticpc	20.8	-	15.5	63.7

“-”: under the detection limit - no element detected

## Data Availability

The original contributions presented in the study are included in the article/Appendix A, further inquiries can be directed to the corresponding author.

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
