# Peer review of "Preliminary Evaluation of Bioactive Collagen–Polyphenol Surface Nanolayers on Titanium Implants: An X-ray Photoelectron Spectroscopy and Bone Implant Study"

_jfb, 2024, doi:10.3390/jfb15070170_

Round 1

Reviewer 1 Report

Comments and Suggestions for Authors

Comments on the paper:

The manuscript reports on the preliminary evaluation of bioactive collagen polyphenol surface nanolayers on titanium implants using X-ray photoelectron spectroscopy and on some in vivo bone implant studies in a rabbit model to test the effects of the surface layer on the conventional osseointegration process.

The topic is considered very relevant to the field of functional materials.

In relation to the research focus, the authors have attempted to answer two fundamental questions that are of great importance for practical clinical application: (a) the actual structure of the collagen-polyphenol surface layer and (b) the effects of the surface layer on the osseointegration process. While in the case of the latter the investigations are quite adequate and the results support the conclusions, in the case of the former some insights are missing. For example, there is no insight into the interaction between the collagen layer and the Ti surface or the collagen/polyphenols/collagen multilayer with the Ti surface to understand the bonding mechanism at the Ti surface. In this sense, the XPS spectrum around the Ti 2p, N 1s and O 1s core level of the Ti implant without coating and with multilayer coating could provide information about the chemical bonding to the Ti surface. Please include the discussion in the text.

The authors provided SEM images of the implant surface in the SI materials: machined and etched. The SEM image of the machined Ticpc and etched Ticpc will be of great value. In addition, the SEM images of Ti with collagen are also very informative.

The SI figure captions are missing.

In general, the present investigations can be considered as a very valuable at upgrading the surface chemistry of titanium implants with the special emphasis on anti-inflammatory and antimicrobial properties.

Author Response

Response to the Reviewers’ comments

We wish to thank all the Reviewers for their very fruitful suggestions to improve the quality of the paper.

The response is organized as follows:

Bold text = Referee’s comments;

Normal text = Authors’ response.

All the corrections or integrations were noted in the main text using Track change.

Reviewer(s)' Comments to Author:

Reviewer #1

In relation to the research focus, the authors have attempted to answer two fundamental questions that are of great importance for practical clinical application: (a) the actual structure of the collagen-polyphenol surface layer and (b) the effects of the surface layer on the osseointegration process. While in the case of the latter the investigations are quite adequate and the results support the conclusions, in the case of the former some insights are missing. For example, there is no insight into the interaction between the collagen layer and the Ti surface or the collagen/polyphenols/collagen multilayer with the Ti surface to understand the bonding mechanism at the Ti surface. In this sense, the XPS spectrum around the Ti 2p, N 1s and O 1s core level of the Ti implant without coating and with multilayer coating could provide information about the chemical bonding to the Ti surface. Please include the discussion in the text.

The Reviewer observation is correct. We were focused on presenting results and overlooked to provide some more basis on the bonding mechanism of the biomolecular layer to the Ti surface. Rightfully, the informed reader can raise the question: nice, but how does collagen bond to the Ti surface?

In this respect, the experimental approach to collagen bonding to Ti  and ensuing multilayer formation on Ti is based on the general phenomenon of irreversible adsorption of proteins (collagen included) at the water/solid interface. See for instance (a) W Norde, F MacRitchie, G Nowicka, J Lyklema, Protein adsorption at solid-liquid interfaces: Reversibility and conformation aspects, Journal of Colloid and Interface Science, Volume 112, Issue 2, 1986; (b) W Norde, J Lyklema,. Why proteins prefer interfaces. Journal of Biomaterials Science, Polymer Edition, 2(3), 183–202, 1991 https://doi.org/10.1080/09205063.1991.9756659).

Physical adsorption of collagen from aqueous solutions to Ti surfaces, as an alternative to chemical linking of collagen to Ti surfaces through spacers or surface functionalities is a widely diffused approach. For a review see for instance: Morra M. Biochemical modification of titanium surfaces: peptides and ECM proteins. Eur Cell Mater. 2006 Jul 24;12:1-15. doi: 10.22203/ecm.v012a01. PMID: 16865661 and the interesting paper C Stewart, B Akhavan, S G Wise, M M.M. Bilek, A review of biomimetic surface functionalization for bone-integrating orthopedic implants: Mechanisms, current approaches, and future directions, Progress in Materials Science, Volume 106, 2019, 100588, ISSN 0079-6425, https://doi.org/10.1016/j.pmatsci.2019.100588.

Following the Reviewer indication, we reworded both the Experimental and the discussion section, to clarify the mechanism of bonding. Three references have been added

The authors provided SEM images of the implant surface in the SI materials: machined and etched. The SEM image of the machined Ticpc and etched Ticpc will be of great value. In addition, the SEM images of Ti with collagen are also very informative.

Good point again. We preferred not to include SEM images of machined Ticpc and etched Ticpc because due to the average roughness of both (Sa > 0,3 micrometers in both cases) and the vertical resolution of the technique, no clear difference is detected through SEM observation with respect to the uncoated ones.

However, we added to the SI AFM images of machined Ti and machined Ticpc compared with etched Ti and etched Ticpc.

The SI figure captions are missing.

Addressed

Reviewer 2 Report

Comments and Suggestions for Authors

The study on bioactive collagen-polyphenol surface nanolayers on titanium implants presents interesting findings, yet significant revisions are necessary to strengthen its scientific merit.

One notable concern is the lack of sample size calculation for the in vivo design, which evaluated four different implants under varying conditions. the allocation and randomization are missing. Additionally, the study fails to mention adherence to ARRIVE guidelines, and the absence of an attached ARRIVE checklist is noted.

Furthermore, to improve clarity and comprehensiveness, the authors should provide detailed information about the implants used, including dimensions, and include a photograph showcasing the design and overall placement of the implants.

Regarding morphometric analysis, a more thorough description and additional information on the regions of interest (ROI) are warranted. Notably, given the bicortical nature of the tibia, it's imperative to ensure adequate bone-to-implant contact (BIC) at both cortical and medullary levels. However, this aspect is not addressed in the current study.

Addressing these concerns would significantly enhance the study's rigor and contribute to its scientific validity.

Comments on the Quality of English Language

The English quality is good; however, professional editing would be recommended. 

Author Response

Response to the Reviewers’ comments

We wish to thank all the Reviewers for their very fruitful suggestions to improve the quality of the paper.

The response is organized as follows:

Bold text = Referee’s comments;

Normal text = Authors’ response.

All the corrections or integrations were noted in the main text using Track change.

Reviewer(s)' Comments to Author:

Reviewer #2 

One notable concern is the lack of sample size calculation for the in vivo design, which evaluated four different implants under varying conditions. the allocation and randomization are missing. Additionally, the study fails to mention adherence to ARRIVE guidelines, and the absence of an attached ARRIVE checklist is noted.

We thank the Reviewer for her/his valuable suggestions, which have significantly enhanced the scientific soundness and validity of our work. According to European Directive and subsequently Italian law, any study involving the use of animals for scientific purpose requires the preparation of a specific Project.

This Project must first be submitted to the Animal Welfare Committee of respective Institution and then to the Ministry of Health for technical and scientific evaluations. A key requirement of the Project, addressing the Reduction aspect of the 3R Principles, is the Statistical Consideration to use the minimum number of animals necessary to meet the research project's objectives. This is achieved through a priori power analysis, which provides guidance on the minimum number of animals needed to detect the hypothesized effect sizes.

Therefore, for this project, the result of the a priori power analysis (G*Power software) provided a sample size of n=12 animals, including 6 per experimental time point. For the determination of the effect size, we relied on previous histomorphometric results obtained by our group [Sartori M, Giavaresi G, Parrilli A, Ferrari A, Aldini NN, Morra M, Cassinelli C, Bollati D, Fini M. Type I collagen coating stimulates bone regeneration and osseointegration of titanium implants in the osteopenic rat. Int Orthop. 2015;39(10):2041-52 - Morra M, Cassinelli C, Cascardo G, Mazzucco L, Borzini P, Fini M, Giavaresi G, Giardino R. Collagen I-coated titanium surfaces: mesenchymal cell adhesion and in vivo evaluation in trabecular bone implants. J Biomed Mater Res A. 2006;78(3):449-58].

Specifically, an effect size of f ≥ 0.50 was assumed, and the sample size was calculated based on a two-way ANOVA model (type of implanted device and experimental time), with a power 1-β=0.80 and a type I error α=0.05.

To address the reviewer's request, the following sentence has been added at the beginning of paragraph 2.5, "In Vivo Study": According to the priori power analysis calculated using the G*Power software, the sample size for the study was n=12 animals, with 6 per experimental time point (2 and 4 weeks). An effect size of f ≥ 0.50 was hypothesized for the quantitative histomorphometric evaluations based on previous results [34,35] in a two-way ANOVA model (device and experimental time), with 1-β=0.80 and α=0.05.

To comply with the ARRIVE guidelines, additional details have been incorporated throughout the manuscript to address the essential items requested.

Furthermore, to improve clarity and comprehensiveness, the authors should provide detailed information about the implants used, including dimensions, and include a photograph showcasing the design and overall placement of the implants.

We thank the Reviewer once again for her/his valuable suggestions, which have significantly improved our work. The dimensions of the implants are reported in the Materials section. 

As noted, it can be difficult to understand the exact positioning of the screws and where the histomorphometric measurements were taken from the reported histological images.

Therefore, to address this, additional figures were added to the manuscript (new Figure 2) to show the reader the implant site selected for the evaluation of the surfaces and the ROI where exactly the histomorphometric measurements were performed. An explanatory caption was also included with the image to help the reader understand our adopted methodology. 

Regarding morphometric analysis, a more thorough description and additional information on the regions of interest (ROI) are warranted. Notably, given the bicortical nature of the tibia, it's imperative to ensure adequate bone-to-implant contact (BIC) at both cortical and medullary levels. However, this aspect is not addressed in the current study.

 The reviewer’s concern is absolutely valid. Providing a supporting image, as added in the new Figure 2, clarifies the site of implantation. As show in new Figure 2, materials have been implanted in the epiphyseal region of the femur and tibia (not in diaphysis), where the cortical bone thickness is significantly lower compared to the diaphysis. Indeed, to effectively evaluate the performance at the cortical level, the implants should be placed at the diaphyseal level to ensure greater contact with the cortical component.

The purpose of this first work was to understand the overall tissue response, not only of the bone tissue but also of the medullary stromal hematopoietic component, in order to highlight any inflammatory reactions evoked by the presence of the collagen/polyphenolic nanolayer coating. Nevertheless, as pointed out by the reviewer, the implant location influence osseointegration due to different bone features from the epiphysis to the diaphysis. Therefore, we have added some sentences in the discussion section to highlight the importance of considering the cortical component in future investigation.

Finally, some of the requests posed by the Reviewer were already addressed in the integration made to the text in response to the previous query.

Reviewer 3 Report

Comments and Suggestions for Authors

The authors are requested to add references to section 2.3 and refrain from using any abbreviations.  Please elaborate if the term (Fibrillation of titanium surface ) is a term introduced by the authors or is quoted from previous research.  If quoted please add suitable references.

The authors are requested to compare their method of collagen application to previously published data "Coating of titanium implants with collagen, RGD peptide and chondroitin sulfate."

Comments on the Quality of English Language

Minor revision is needed

Author Response

Response to the Reviewers’ comments

We wish to thank all the Reviewers for their very fruitful suggestions to improve the quality of the paper.

The response is organized as follows:

Bold text = Referee’s comments;

Normal text = Authors’ response.

All the corrections or integrations were noted in the main text using Track change.

Reviewer(s)' Comments to Author:

Reviewer #3

The authors are requested to add references to section 2.3 and refrain from using any abbreviations.  Please elaborate if the term (Fibrillation of titanium surface) is a term introduced by the authors or is quoted from previous research.  If quoted please add suitable references.

The wording of sample preparation has been changed to address the admittedly unclear sentence: “Fibrillation of titanium surface. It is actually collagen that is present in fibrillary form, due to the pH of the final solution, as per common practice

The authors are requested to compare their method of collagen application to previously published data "Coating of titanium implants with collagen, RGD peptide and chondroitin sulfate”

The methods of application of collagenbased coatings, including those involving RGD peptides and chondroitin sulphate, are reviewed in many published papers, included new reported quotation in the present ms. Collagen-chondroitin sulphate coating is an exceedingly interesting topic, with reported excellent results even in osteopenic animal models. 

Conceptually, the two approaches are very similar, except that Collagen-CS is based on molecules from animal sources, Collagen-polyphenols exploits molecules both from the animal and plant kingdom.

Even if nicely to the point when it comes to the general field of application, we feel that the Reviewer request is a bit outside the scope of the present work that explicitly aims at unravelling some key aspect of the present method, as stated in the introduction.

Round 2

Reviewer 1 Report

Comments and Suggestions for Authors

Thank to the authors for taking into account the given suggestions and making certain corrections.

The only thing I would suggest is to correct the added sentence in line 452. "Irreversible adsorption of preteinaceous Collagen at the liquid/solid interface ..."  into "Irreversible adsorbed proteinaceous collagen at the liquid/solid interface [39-41] was served both as a primer to enhance adhesion of phenolics to titanium and because of its known effects on tissue healing."

Author Response

Response to the Reviewers’ comments

We wish to thank all the Reviewers for their very fruitful suggestions to improve the quality of the paper.

The response is organized as follows:

Bold text = Referee’s comments;

Normal text = Authors’ response.

All the corrections or integrations were noted in the main text using Track change.

Reviewer(s)' Comments to Author:

Reviewer #1

Thank to the authors for taking into account the given suggestions and making certain corrections.

The only thing I would suggest is to correct the added sentence in line 452. "Irreversible adsorption of preteinaceous Collagen at the liquid/solid interface ..."  into "Irreversible adsorbed proteinaceous collagen at the liquid/solid interface [39-41] was served both as a primer to enhance adhesion of phenolics to titanium and because of its known effects on tissue healing."

We would like to thank the Reviewer for her/his efforts in improving our work.As suggested the sentence in line 452 has been corrected.

Reviewer 2 Report

Comments and Suggestions for Authors

The authors have improved the manuscript; however, further enhancements are needed. In Figure 2, I suggest revising the legend to be more concise and informative. Additionally, the scale bar is not visible in the figures, and the image quality needs improvement. The histomorphometric measurements were taken only in the medullary zone, excluding the cortical zone. I recommend including measurements from both zones for a more comprehensive analysis.

Comments on the Quality of English Language

Some minor typos in the discussion need to be corrected. 

Author Response

Response to the Reviewers’ comments

We wish to thank all the Reviewers for their very fruitful suggestions to improve the quality of the paper.

The response is organized as follows:

Bold text = Referee’s comments;

Normal text = Authors’ response.

All the corrections or integrations were noted in the main text using Track change.

Reviewer(s)' Comments to Author:

Reviewer #2

The authors have improved the manuscript; however, further enhancements are needed. In Figure 2, I suggest revising the legend to be more concise and informative. Additionally, the scale bar is not visible in the figures, and the image quality needs improvement. The histomorphometric measurements were taken only in the medullary zone, excluding the cortical zone. I recommend including measurements from both zones for a more comprehensive analysis.

We would like to thank the Reviewer for her/his efforts in improving our work. As suggested, we revised the caption for Figure 2 to be more concise and informative. We also highlighted the scale bar in each image and improved the quality of the final image.

Regarding histomorphometric measurements, as shown in Figure 2, the implant of the experimental surfaces was performed at the epiphyseal level of the femur and tibia where cancellous (spongy) bone is the main component which is covered by a thin layer of compact (cortical) bone. Veterinary anatomical studies indicate that the cortical thickness of the epiphysis in rabbits naturally varies based on age, breed, and health, but it is typically between 0.2-0.5 mm in the proximal epiphyses of the femur and tibia. Furthermore, due to the very short investigated experimental times (2 and 4 weeks), the cortical bone layer had not yet fully reformed as it was affected by the both surgical procedure and the subsequent periosteal reaction.

For this reason, in the first round of revision, we added the following sentences to the discussion section: "However, our observations and measurements focused exclusively on the response of the trabecular bone tissue. Further studies are needed to understand the response at the cortical bone level, as different features characterize the structure of trabecular and cortical bone," as we are aware that our results only refer to the evaluation derived from trabecular bone.

Manuscript has been deeply revised and typos corrected.
